# SparseLGS: Fast Language Gaussian Splatting from Sparse Multi-View Images

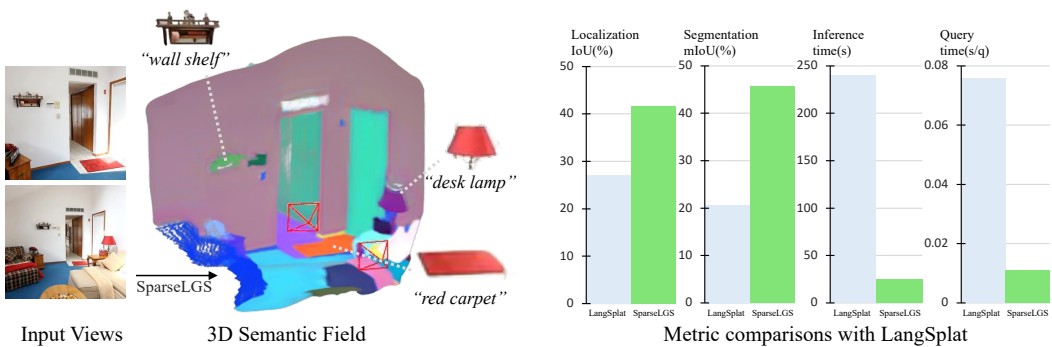

Figure 1: With just two RGB views, our method infers 3D semantic fields in under 30 seconds without per-scene optimization. On LERF and 3D-OVS datasets (image resolution 416 × 576), query time is 0.011 seconds/query, outperforming existing methods in both speed and IoU metrics.

## Abstract

3D semantic field learning is crucial for applications like autonomous navigation, AR/VR, and robotics, where accurate comprehension of 3D scenes from limited viewpoints is essential. Existing methods struggle under sparse view conditions, relying on inefficient per-scene multi-view optimizations, which are impractical for many real-world tasks. To address this, we propose SparseLGS, a feed-forward method for constructing 3D semantic fields from sparse viewpoints, allowing direct inference of 3DGS-based scenes. By ensuring consistent SAM segmentations through video tracking and using low-dimensional indexing for high-dimensional CLIP features, SparseLGS efficiently embeds language information in 3D space, offering a robust solution for accurate 3D scene understanding under sparse view conditions. In experiments on two-view sparse 3D object querying and segmentation in the LERF and 3D-OVS datasets, SparseLGS outperforms existing methods in chosen IoU, Localization Accuracy, and mIoU. Moreover, our model achieves scene inference in under 30 seconds and open-vocabulary querying in just 0.011 seconds per query.

## 1 Introduction

In many real-world applications, such as autonomous navigation, AR, VR, and robotics, understanding and interacting with 3D environments using natural language is essential. In systems like autonomous driving, where the number of onboard cameras is limited, efficient scene inference is still required. Similarly, AR/VR applications often rely on limited perspectives to support accurate 3D scene understanding. Under these sparse-view conditions, achieving fast and accurate 3D semantic field learning becomes both critical and challenging.

While methods like LERF (Kerr et al., 2023) and LangSplat (Qin et al., 2024) have made significant advances in 3D language scene learning, they rely heavily on dense multi-view information, which makes them unsuitable for scenarios with limited viewpoints. When only a few views are

available, it becomes challenging to reconstruct detailed 3D geometry and maintain consistent language information across those views. This hinders the system's ability to accurately perform 3D language field learning, as the semantic information cannot be reliably integrated across different perspectives. Additionally, existing methods often require per-scene optimization, which is not only time-consuming but impractical for real-world applications, such as robotics, autonomous navigation, or industrial monitoring, where rapid inference is necessary. In such cases, the inability to perform direct inference on 3D scenes limits their utility in providing efficient semantic understanding of the environment.

To overcome these limitations and enable the broader application of semantic fields in real-world scenarios, we focus on improving inference speed and open-vocabulary query quality through three key innovations. First, inspired by recent feed-forward approaches, we can adopt a feed-forward framework that allows for direct inference without the need for per-scene optimization after model convergence, greatly reducing time costs. Additionally, We can leverage 3D Gaussian Splatting (3DGS) (Kerbl et al., 2023) for its fast and photorealistic rendering to accelerate the rendering process in our model. Moreover, to enable open-vocabulary query, related works learn high-dimensional CLIP (Radford et al., 2021) features or use additional networks to reduce the dimensionality of these features, which often hampers inference speed. We can seek more efficient feature embedding methods to overcome this limitation and improve query performance.

Based on these insights, we present the first feed-forward method for constructing semantic fields from sparse viewpoints. Our model takes sparse-view RGB images as input and utilizes two branches: one for predicting base Gaussian parameters and another for semantic parameters, enabling fast construction of 3DGS-based semantic fields. For base Gaussian parameters, we build on existing 3DGS feed-forward methods like (Charatan et al., 2024; Chen et al., 2024), while a dedicated module handles feature extraction and mapping for semantic predictions. After predicting both sets of parameters, we render 3D semantic features using the splatting technique. Then we introduce a multi-view language memory bank with low-dimensional semantic label IDs, which also facilitates efficient matching between rendered semantic features and query prompts. Additionally, we address inconsistent multi-view segmentation from the SAM model by applying video object tracking techniques. In sparse-view experiments on the LERF and 3D-OVS datasets, our method achieves sharp 3D object segmentation boundaries and outperforms existing approaches in chosen IoU, Localization Accuracy, and mIoU. At a resolution of $416 \times 576$, our method delivers scene inference in around 25 seconds and query response times of 0.011 seconds, significantly faster than current methods.

In summary, our contributions are:

- To our knowledge, we are the first to propose a feed-forward method for constructing 3D semantic fields from sparse viewpoints. Our approach generates 3D semantic fields from only two-view images, enabling fast inference with strong generalization capabilities.

- We introduce a multi-view language memory bank that links semantic masks to natural language information, addressing the limitations of existing methods in sparse-view scenarios and significantly accelerating open-vocabulary querying.

- Our method surpasses state-of-the-art approaches in sparse-view 3D object localization and segmentation tasks, achieving notable improvements in chosen IoU, Localization Accuracy, and mIoU, while providing 10× faster scene inference and 7× faster open-vocabulary querying than LangSplat.

## 2 RELATED WORKS

**NeRF and 3D Gaussian Splatting (3DGS).** In recent years, Neural Radiance Fields (NeRF) (Mildenhall et al., 2021) and 3D Gaussian Splatting (3DGS) (Kerbl et al., 2023) have made significant advancements in 3D scene modeling and rendering. NeRF (Mildenhall et al., 2021) synthesizes photorealistic scenes through volumetric rendering and multi-view consistency backpropagation, but its reliance on ray marching results in slow rendering speeds. Subsequent research, such as mip-NeRF (Barron et al., 2021) and TensoRF (Chen et al., 2022), improved its rendering efficiency and multi-resolution representation. Unlike NeRF, 3DGS employs explicit 3D Gaussian distributions instead of MLPs, achieving real-time and efficient rendering through adaptive density control and

tile-based rasterization. Tang et al. (2023), Liu et al. (2024) demonstrated the potential of 3DGS in generative tasks. Methods like (Yang et al., 2024; Wu et al., 2024a; Li et al., 2024; Yang et al., 2023; He et al., 2024) enhanced Gaussians' flexibility in handling dynamic and complex scenes by learning 3D Gaussians in canonical space combined with deformation fields. Zhang et al. (2024) introduces separated intrinsic and dynamic appearance feature to capture the unchanged scene appearance along with dynamic variation like illumination and weather. Additionally, Qin et al. (2024); Wu et al. (2024b); Shi et al. (2024) incorporate language information into 3DGS scene representation, enabling high-quality open-vocabulary 3D scene queries.

**Sparse View Reconstruction.** Although techniques of NeRF and 3DGS perform well in dense-view scenarios, they face challenges in sparse-view settings, particularly in maintaining high-quality scene generation under sparse data conditions, necessitating further optimizations to improve generation quality and efficiency. Methods like (Yu et al., 2021; Charatan et al., 2024; Chen et al., 2024) use a feed-forward manner to learn the 3D scene, which take images as input and predict the parameters of the 3D representation.Specifically, PixelNeRF (Yu et al., 2021) uses convolutional neural networks to extract features from input contexts, addressing the challenges of low-quality reconstructions caused by sparse views. PixelSplat leverages epipolar transformers to extract scene features, while MVSplat uses cost volumes to improve scene feature extraction. FreeNeRF further improves reconstruction by employing frequency and density regularization strategies to reduce artifacts from insufficient inputs, without additional computational costs. To mitigate overfitting in 3DGS (Kerbl et al., 2023) models under sparse views, approaches like FSGS (Zhu et al., 2023) and SparseGS (Xiong et al., 2023) introduce depth estimators to regularize the optimization process. FreeSplat (Wang et al., 2024) builds on this by incorporating not only cost volume construction but also multi-scale feature aggregation, allowing accurate 3D Gaussian distribution localization and supporting free-viewpoint synthesis. These advancements demonstrate the growing potential of sparse-view reconstruction technologies. Learning from the 3DGS-based generative frameworks (Charatan et al., 2024; Chen et al., 2024; Wang et al., 2024), SparseLGS adopts a similar approach to infer high-quality 3DGS semantic scenes in sparse-view semantic learning tasks.

**3D Semantic Field.** Integrating semantic information into 3D scenes is crucial for downstream fields such as robotics, AR, and VR. Zhi et al. (2021) first introduced semantic layers in NeRF, laying the foundation for deeper scene understanding. Subsequent studies, such as LERF (Kerr et al., 2023), ISRF (Goel et al., 2023), and DFF (Yen-Chen et al., 2022), have incorporated more refined semantic segmentation techniques, significantly improving the level of detail in 3D visual representations. Furthermore, 3DGS technology has also made breakthroughs in semantic tasks. The foundational work of Kerbl et al. (2023) and the subsequent developments of EgoLifter (Gu et al., 2024), SAGA (Cen et al., 2023), Gaussian Grouping (Ye et al., 2023), and Gaga (Lyu et al., 2024) have greatly optimized the application of 3DGS in semantic segmentation. LangSplat, LeGaussian, OpenGaussian and FastLGS (Ji et al., 2024) further explored the integration of language information into 3DGS, deepening semantic interpretation in 3D scene modeling. However, these advanced techniques often exhibit limitations in handling sparse view inputs, especially when viewpoints are highly limited, making it difficult to effectively manage semantic consistency between views. The lack of sufficient RGB information leads to suboptimal reconstruction quality of both 3D scenes and their semantic content, highlighting the urgent need for efficient and accurate 3D semantic learning under sparse views.

## 3 OUR APPROACH

In this section, we present a detailed overview of our SparseLGS. As shown in Figure 2, we begin by explaining the **Base Gaussians Prediction** and **Semantic Parameters Prediction**, which enable rapid construction of semantic fields. Following this, we perform semantic splatting based on the predicted semantic Gaussians (Section 3.1). We also explain how high-dimensional language information is embedded into the 3D semantic field using a **Multi-view Language Memory Bank**, allowing for efficient **Open-vocabulary Querying** (Section 3.2 and Section 3.3).

### 3.1 3DGS-BASED SEMANTIC PARAMETERS PREDICTION

**Base Gaussians Prediction.** Inferring the semantic field directly from RGB images makes it difficult for the model to fully capture the scene's geometric structure. A robust semantic field repre-

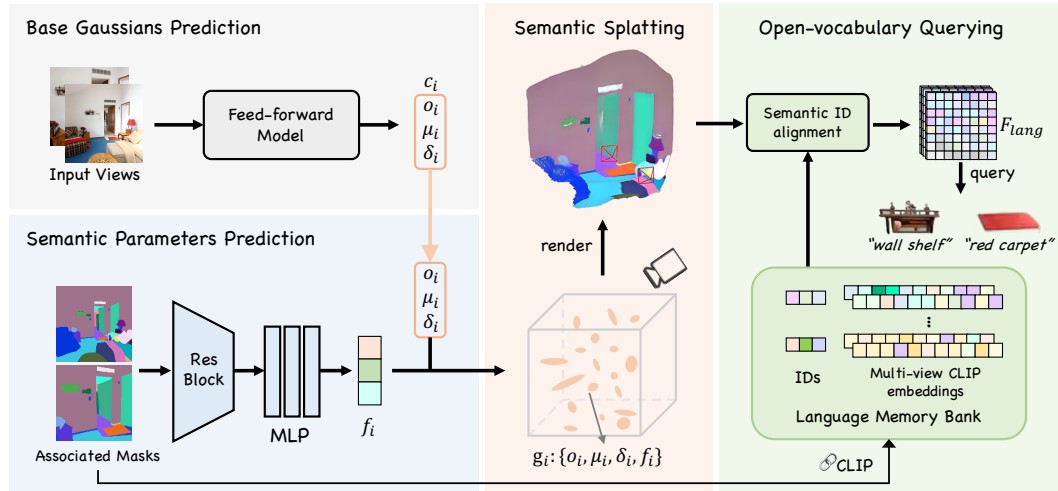

Figure 2: **Overview of our approach.** Starting with two-view RGB images, we first apply SAM (Kirillov et al., 2023) and a video object tracking model to generate consistent labeled masks (Associated Masks) and construct the language memory bank to store CLIP embeddings (Radford et al., 2021) for each label ID (Section 3.2). The RGB images and associated masks are processed by the **Base Gaussians Prediction** and **Semantic Parameters Prediction** modules, which are then combined to generate semantic Gaussians. These are then used to render the 3D semantic field (Section 3.1). Finally, we perform **Open-vocabulary Querying** through our **Multi-view Language Memory Bank** (Section 3.2 and Section 3.3), enabling efficient querying of objects within the 3D scene.

sentation requires a deep understanding of the scene's geometry. To address this, we first apply 3D scene reconstruction methods to provide the model with a foundational grasp of the scene, which then guides the learning of semantic features. As a result, we construct a regular color field using a feed-forward 3D Gaussian Splatting (3DGS) (Kerbl et al., 2023) method to derive the base Gaussians:

$$\{c_i, o_i, \mu_i, \delta_i\} = \text{GS-Pred}\left\{\left\{R_j^{3 \times H \times W}\right\}\right\} \tag{1}$$

here, $c_i$, $o_i$, $\mu_i$, and $\delta_i$ represent the color, opacity, mean, and covariance of the $i$-th Gaussian, respectively. The term $R_j^{3 \times H \times W}$ refers to the RGB image from the $j$-th view, where $H$ and $W$ correspond to the height and width of the image. It is important to note that in this module, we have the flexibility to use any feed-forward model. In our work, we utilize an established method (Chen et al., 2024) to predict the base Gaussians. These base Gaussian parameters are essential for accurately representing both the scene's geometry and appearance, directly affecting the quality of semantic field reconstruction. The number of Gaussian splats in this method is determined by the input views and the image resolution, ensuring that each pixel is associated with a corresponding Gaussian point. As a result, the prediction of base Gaussians plays a crucial role in structuring the scene representation. We can either train a complete model ourselves or use pre-trained model weights from existing approaches. Once trained, the feed-forward model in this module is frozen to maintain stability for subsequent semantic field learning.

**Semantic Parameters Prediction.** Besides predicting the base Gaussian parameters, we also predict the 3DGS semantic parameters in another branch. First, we process the input RGB images from different viewpoints to obtain consistent 2D segmentation maps across views, denoted as $\{S_i^{1 \times H \times W}\}$, where $H$ and $W$ represent the height and width of the image, respectively. Then, during the construction of the language memory bank, which will be detailed later, we map the segmentation maps into semantic label images $\{M_i^{3 \times H \times W}\}$. The dimensional extension to 3 is necessary because a 1D index space cannot effectively separate the semantic labels representing different objects within the scene. Next, we apply a simple CNN network to extract features from $M_i$. After being input into the network, $M_i$ first passes through a convolutional layer and then through multiple downsampling operations based on Residual Blocks (He et al., 2016). Finally, it is up-

sampled back to the original resolution and added to the original input $M_i$. The resulting feature map is flattened and then passed through an MLP, which maps it to the Gaussian semantic feature parameters $f_i$. These parameters are 3-dimensional, corresponding directly to the 3D labels of $M_i$. Finally, $f_i$ is combined with the previously obtained base Gaussian parameters $o_i, \mu_i, \delta_i$ to generate the final 3D semantic field representation $\{o_i, \mu_i, \delta_i, f_i\}$.

**Semantic Splatting.** 3DGS explicitly represents a 3D scene as a collection of anisotropic 3D Gaussians, with each Gaussian $G(x)$ characterized by a mean $\mu \in \mathbb{R}^3$ and a covariance matrix $\Sigma$:

$$G(x) = \exp\left(-\frac{1}{2}(x - \mu)^\top \Sigma^{-1}(x - \mu)\right). \tag{2}$$

and based on the Gaussian representation $\{o_i, \mu_i, \delta_i, f_i\}$ obtained, our semantic feature rendering $F(v)$ for pixel $v$ is defined as:

$$F(v) = \sum_{i \in \mathcal{N}} f_i \alpha_i \prod_{j=1}^{i-1} (1 - \alpha_j) \tag{3}$$

here, $C(v)$ represents the rendered color at pixel $v$ for a specific view, and $\mathcal{N}$ denotes the number of Gaussians in the tile. $\alpha_i = o_i G_i^{2D}(v)$, where $G_i^{2D}(\cdot)$ refers to the projection of the $i$-th Gaussian onto the 2D image plane.

### 3.2 Multi-view Language Memory Bank

**Mask Association.** Directly using the multi-view segmentation results inferred by the SAM model as input for our model can lead to confusion, as the inconsistent 2D segmentations across views interfere with the spatial mapping of semantic labels, as shown in the example from the appendix (Figure 10). Existing methods mitigate this issue by using dense viewpoint information and multi-scale segmentation of SAM results. However, when reduced to the extreme sparse scenario, especially when only two views are available, this multi-scale approach to SAM segmentation becomes ineffective. Research on Gaussian scene segmentation (Ye et al., 2023; Dou et al., 2024) treats multi-view images as consecutive frames in a video and applies a video object tracking technique (Cheng et al., 2023) to unify the multi-view 2D segmentation masks inferred by SAM, achieving relatively good results. We adopted this method to enforce multi-view consistency on SAM's inference results. However, when only two sparse-view images are directly processed by the video tracking model, the lack of sufficient reference views often result in poor segmentation consistency. To ensure more stable processing results for sparse-view image inputs, we replicated the input images five times to create a 10-frame sequence and processed it using the model, resulting in a set of 10 segmentation images with matched indices: $\{T_{11}, T_{12}, \ldots, T_{15}, T_{21}, T_{22}, \ldots, T_{25}\}$, where $T_{ij}$ represents the $j$-th frame segmentation corresponding to the $i$-th view. Each pixel in the segmentation images is labeled with the corresponding object index. For each pixel $v$ in the $i$-th view, we choose the category through a voting algorithm:

$$S_i(v) = \underset{x \in \{T_{ij}(v)\}}{\arg\max} \, \text{count}(x) \tag{4}$$

thus, we obtained a set of mask-matched results with improved consistency $\{S_i^{1 \times H \times W}\}$, where $H$ and $W$ represent the height and width of the segmentation image, respectively.

**Construction of Language Memory Bank.** Based on the multi-view consistent segmentation obtained, we use CLIP to encode the corresponding regions of the RGB images, resulting in high-dimensional language features $L_{i1}$ and $L_{i2}$ (512 dimensions in our experiments) for the two views. If the object does not exist in a particular view, the corresponding language feature is represented by a zero vector. Directly applying these high-dimensional language features to 3DGS would lead to a memory overflow in 3DGS. Qin et al. (2024) addresses this issue by using a per-scene autoencoder to compress the high-dimensional features, which are later decoded during querying. However, this approach is not suitable for our method, as our goal is to infer the semantic field for new scenes without retraining. A simple autoencoder cannot effectively fit the wide range of language information in the real world. Additionally, the differences in language information between the two views might cause the inferred language features to neutralize, making it impossible to recover the corresponding semantics. Therefore, this dimensionality reduction method is not applicable.

We adopt a method that uses evenly distributed vector IDs in a low-dimensional space (set to 3 dimensions in our experiments) to represent the multi-view high-dimensional language features of specific objects. Based on the total number of object categories across all views, $N$, we partition the 3D space $[0,1]^3$ into approximately $\lceil\sqrt[3]{N}\rceil^3$ evenly spaced vectors, from which we randomly select $N$ vectors to serve as IDs. These IDs are then matched with the corresponding object mask indices in $\{S_i^{1\times H\times W}\}$, converting them into semantic label maps $\{M_i^{3\times H\times W}\}$. The reason we do not directly use 1D object mask labels as semantic labels is that the 1D space is too low-dimensional to effectively separate the labels representing different objects in the scene. Based on the above discussion, we construct the multi-view language feature memory bank:

$$\{\text{ID} : \{L_{i1}, L_{i2}\}\} \tag{5}$$

for the semantic label prediction result $F(v)$ rendered from the semantic field for a specific pixel $v$, the ID closest in L1 distance is selected to match the corresponding $\{L_{i1}, L_{i2}\}$.

### 3.3 Open-vocabulary Querying

Based on the previously mentioned multi-view language memory bank, we can easily map all pixels $\{v\}$ in the rendered view $I$ to the corresponding multi-view CLIP language encodings $\{L_1^v, L_2^v\}_v$. Simultaneously, for open-vocabulary query text $qry$, we can also obtain its language encoding $\phi_{qry} = \text{CLIP}(qry)$. Then we calculate the relevance score between $\text{CLIP}(qry)$ and the multi-view language features $\{L_1^v, L_2^v\}_v$ for each pixel $v$, using the following formula:

$$S_{relevancy} = \left\{ \max_j \min_i \frac{\exp(L_j^v \cdot \phi_{qry})}{\exp(L_j^v \cdot \phi_{qry}) + \exp(L_j^v \cdot \phi_{\text{canon}}^i)} \right\}_v \tag{6}$$

where $\phi_{\text{canon}}^i$ represents the CLIP encoding of predefined canonical phrases (such as "object", "things", "stuff", and "texture").

For each open-vocabulary 3D object query, we first render the semantic label feature map for the corresponding view. Then, we map it to a multi-view high-dimensional language feature map using the multi-view language memory bank. Next, we compute the relevancy map between these features and the query text. We select all pixel regions in the relevancy map with scores higher than a threshold $n$ as the object query result, and the pixel with the highest relevancy score is identified as the object localization result. For the 3D scene segmentation task, we compare the relevancy scores of each pixel with multiple query texts and assign the category of the query text with the highest score to that pixel.

## 4 Experiments

### 4.1 Implementation Details

For the training process, we conducted 80,000 iterations on two NVIDIA RTX 3090 GPUs with a batch size of 1. We choose to use MVSplat (Chen et al., 2024) as our base model in the base Gaussian prediction module. The base model will be frozen after pre-trained weights were loaded, then we will focus on optimizing the semantic parameters prediction branch. We used a subset of the RealEstate10K dataset (Zhou et al., 2018), which originally consists of video frames extracted from YouTube videos, with corresponding camera poses obtained through COLMAP by the authors of PixelSplat (Charatan et al., 2024). Following their setup, we resized all images to 256 × 256 resolution when training on the processed RealEstate10K subset containing multi-view mask segmentations. These multi-view masks, generated via video tracking and the SAM model (Cheng et al., 2023; Kirillov et al., 2023), provided spatially consistent supervisory signals. We computed a mean squared error (MSE) loss between the predicted semantic feature maps and the ground truth mask identifiers during training.

For inference, we provided two-view RGB images and their corresponding camera poses, skipping further retraining on datasets such as LERF (Kerr et al., 2023) and 3D-OVS (Liu et al., 2023). Instead, we directly input two images and their camera poses from these datasets, using our model to predict the semantic-encoded features. In these inference experiments, we standardized the input image resolution to 416 × 576 for LERF and 3D-OVS. Our experiments on two RTX 3090 GPUs,

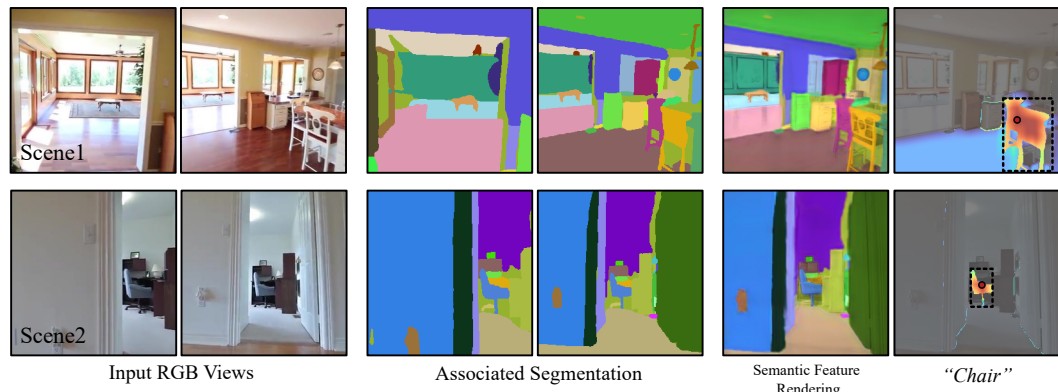

Input RGB Views      Associated Segmentation      Semantic Feature Rendering      *"Chair"*

Figure 3: **Visualization of RealEstate10K dataset scene.** The images depict two scenes from the test set. On the far right, the results show the object localization when querying with the word "Chair".

each utilizing around 9GB of VRAM, demonstrated that scene inference with $416 \times 576$ resolution images takes approximately 0.3 seconds, while single-view rendering takes around 0.0006 seconds. Additionally, model inference on a single 3090 GPU consumes roughly 6GB of VRAM.

## 4.2 RESULTS ON REALESTATE10K DATASET

We used a subset extracted from the RealEstate10K dataset, consisting of 50 files representing different scenes. The scene images were processed sequentially using the SAM model, video tracking, and the CLIP model to create the semantic dataset for training our model. We froze the weights of the MVSplat model and only optimized the parameters of the semantic parameters prediction module. With a batch size of 1, the model was trained over 80,000 iterations on two NVIDIA RTX 3090 GPUs. We manually annotated some new scenes for testing, and the visualization results are shown in Figure 3. It can be observed that our model demonstrates good generalization capability for semantic scene inference in novel scenes.

We also evaluated the performance of using only the MVSplat model and our semantic parameters prediction module, removing mask matching and the multi-view language memory bank. In this setup, we used the SAM model for 2D segmentation and relied on the autoencoder from LangSplat to encode and decode language features. The results, shown in Figure 10 in the appendix, demonstrate that for inconsistent 2D segmentation inputs, the predicted Gaussian semantic parameters are highly disorganized. The rendered semantic features for the original input viewpoints, as well as the correlation heatmaps, are extremely chaotic, leading to incorrect results for novel view 3D queries.

|  | figurines | ramen | teatime | waldo kitchen | overall |
|---|---|---|---|---|---|
| LERF | 20.88 | 16.06 | 21.64 | 15.12 | 18.43 |
| LangSplat | 43.05 | 47.66 | 8.70 | 8.94 | 27.09 |
| SparseLGS | **47.28** | **55.21** | **40.65** | **22.86** | **41.50** |

Table 1: Quantitative comparisons of 3D semantic segmentation on LERF dataset. We report the average IoU scores (%).

## 4.3 COMPARISONS ON LERF DATASET

**Quantitative Results.** We compared our method to LERF and LangSplat on the task of 3D object open-vocabulary query localization using the LERF dataset. The chosen IoU scores (Table 1) show that our method consistently outperforms both LERF and LangSplat across all scenes, with a significantly higher overall average IoU. For object localization accuracy (Table 2), which checks whether the predicted point lies within the correct object, our approach also excels, especially in scenes like

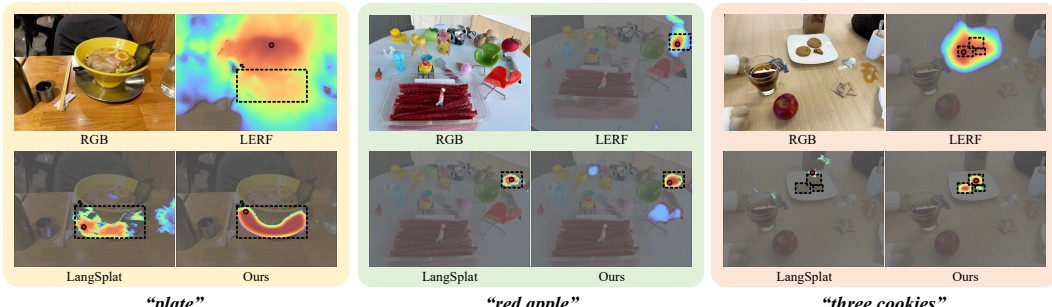

Figure 4: Qualitative comparisons of open-vocabulary 3D object localization on the LERF dataset. The red points are the model predictions and the black dashed bounding boxes denote the annotations.

|  | figurines | ramen | teatime | waldo kitchen | overall |
|---|---|---|---|---|---|
| LERF | 58.33 | 40.00 | **62.50** | 46.15 | 51.75 |
| LangSplat | 83.33 | **90.00** | 18.75 | 15.38 | 51.87 |
| SparseLGS | **91.67** | 80.00 | 50.00 | **61.54** | **70.80** |

Table 2: Localization accuracy (%) comparisons on LERF dataset.

"figurines" and "waldo kitchen." The only slight underperformance is in "ramen" and "teatime," where the localization accuracy is marginally lower than the other methods.

**Visualization Results.** Figure 4 visualizes the 3D object query results from various scenes in the LERF dataset. In the "plate" query, LERF over-associates much of the ramen scene with "plate," while LangSplat, despite identifying the object, produces incomplete results due to poor geometric consistency. Our method delivers a more complete and accurate result. For the "three cookies" query, LangSplat shows minimal relevance, and LERF incorrectly selects part of the plate. Our method, however, precisely localizes the cookies. Similarly, in the "red apple" query, LERF and LangSplat face challenges, while our method, despite slightly highlighting both a green apple and a red chair due to the dual nature of "red" and "apple" in the query, performs better overall.

## 4.4 COMPARISONS ON 3D-OVS DATASET

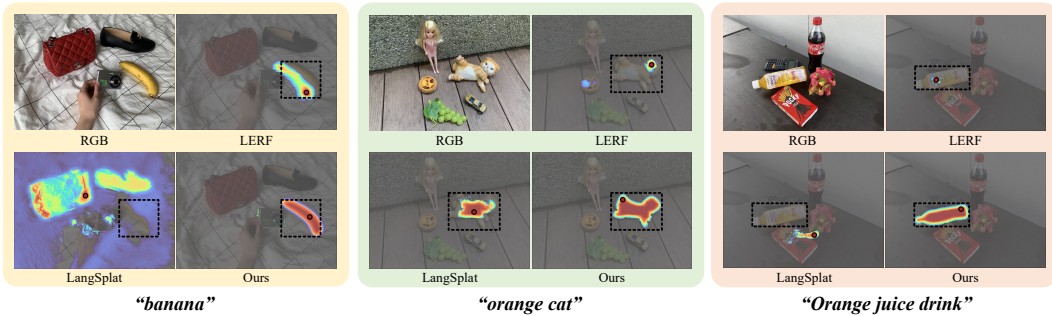

Figure 5: Qualitative comparisons of open-vocabulary 3D object localization on the 3D-OVS dataset. The red points are the model predictions and the black dashed bounding boxes denote the annotations.

**Quantitative Results.** We compared our method with LERF, 3D-OVS, and LangSplat on the 3D-OVS dataset for 3D object localization and segmentation tasks. Results across ten scenes are shown in Figure 3 and Figure 4, with additional results in Appendix Figure 7 and Figure 8. Our method consistently outperforms others, nearly doubling the average IoU for single open-vocabulary object

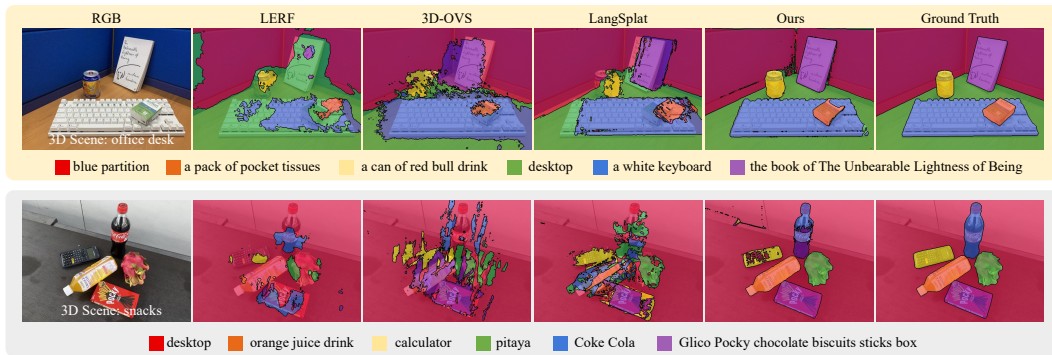

Figure 6: Qualitative comparisons of open-vocabulary 3D object segmentation on the 3D-OVS dataset.

|  | bed | bench | blue sofa | covered desk | lawn | overall |
|---|---|---|---|---|---|---|
| LERF | 20.85 | 26.51 | 14.21 | 7.57 | 0.00 | 13.83 |
| LangSplat | 0.62 | 33.31 | 19.46 | 15.51 | 30.40 | 19.86 |
| SparseLGS | **66.60** | **81.35** | **50.48** | **51.97** | **60.70** | **62.22** |

Table 3: Quantitative comparisons of 3D semantic segmentation on 3D-OVS dataset. We report the average IoU scores (%). The metrics for the remaining scenes can be found in the appendix Table 7.

queries. In multi-object segmentation tasks, measured by mIoU, we achieve superior results in most scenes, with the overall mIoU significantly surpassing existing methods.

**Visualization Results.** Figure 5 compares object localization results on the 3D-OVS dataset. Both our method and LERF accurately locate target objects, but our method provides more complete object queries, while LERF only ensures the localization point is on the object. Our method shows stronger correlation between the object and query text. LangSplat fails in the queries for "banana" and "orange juice drink" and misses key parts of the "orange cat" doll. Figure 6 compares the 3D object segmentation results for the "office desk" and "snacks" scenes. Other methods show misaligned or chaotic segmentation under sparse views, while our method achieves more accurate segmentation of objects and background. Additional results are provided in the appendix.

## 4.5 ABLATION STUDY

**Ablation study on LERF dataset.** The results of the ablation study on the LERF dataset (Table 5) show a significant improvement in Chosen IoU (object region accuracy) and Localization Accuracy as model components are gradually introduced. Without the feed-forward Gaussian inference model, mask association (MA), or Multi-View Language Memory Bank (MV-LMB)—essentially replicating LangSplat—the model achieves 27.09% Chosen IoU and 51.87% Localization Accuracy. However, when only the feed-forward model is introduced, both metrics drop sharply to 5.06% and 9.11%, indicating the detrimental effect of inconsistent 2D semantic feature supervision across views. Adding mask association (MA) further decreases performance (3.06% Chosen IoU and 5.05% Localization Accuracy), showing that mask association alone is insufficient under sparse views. Finally, introducing the MV-LMB boosts Chosen IoU to 41.50% and Localization Accuracy to 70.80%, demonstrating the significant role of consistent multi-view language features in improving inference and query performance.

**Comparison between our method and the 2D query method** Our approach renders language-encoded feature maps from novel viewpoints and compares them with the Multi-View Language Memory Bank, while the 2D method renders RGB images, segments them with SAM, and encodes regions using CLIP for comparison with query text. Table 6 shows the LERF dataset results, comparing Chosen IoU, Localization Accuracy, and Query Speed. Both methods perform similarly in terms of IoU and accuracy, but our method stands out with near real-time querying, requiring only 0.011 seconds per query, whereas the 2D method takes up to 18 seconds for a single query.

|  | bed | bench | blue sofa | covered desk | lawn | overall |
|---|---|---|---|---|---|---|
| LERF | 12.67 | 29.92 | 23.65 | 20.63 | 5.50 | 18.47 |
| 3D-OVS | 15.86 | 17.16 | 23.41 | 18.13 | 13.70 | 17.65 |
| LangSplat | 1.72 | 27.08 | 6.23 | 21.17 | 31.69 | 17.58 |
| SparseLGS | **41.33** | **47.02** | **30.25** | **47.67** | **54.38** | **44.13** |

Table 4: Quantitative comparisons of 3D semantic segmentation on 3D-OVS dataset. We report the average mIoU scores (%). The metrics for the remaining scenes can be found in the appendix Table 8.

| Component | | | Performance | |
|---|---|---|---|---|
| Feed-forward Model | MA | MV-LMB | Chosen IoU (%) | Localization Accuracy (%) |
|  |  |  | 27.09 | 51.87 |
| ✓ |  |  | 5.06 | 9.11 |
| ✓ | ✓ |  | 3.06 | 5.05 |
| ✓ | ✓ | ✓ | **41.50** | **70.80** |

Table 5: The ablation study on the LERF dataset reports Chosen IoU (accuracy of selected object regions) and Localization Accuracy (precision of object localization). The configurations tested include whether the Gaussian feed-forward inference model was used, indicated as Feed-forward Model; whether mask association was performed (MA); and whether the Multi-View Language Memory Bank was performed (MV-LMB).

| Novel Viewpoint Query Method | Chosen IoU (%) | Localization Accuracy (%) | Speed (s/q) |
|---|---|---|---|
| 2D Query | **44.64** | 65.08 | 18 |
| SparseLGS | 41.50 | **70.80** | **0.011** |

Table 6: Comparison of SparseLGS and 2D query method for novel viewpoint object query on the LERF dataset (416 x 576 resolution).

## 5 LIMITATIONS AND CONCLUSION

**Conclusion.** We present SparseLGS, a feed-forward method for constructing 3D semantic fields from sparse viewpoints. By leveraging a feed-forward framework for base Gaussians and semantic parameters prediction, along with a multi-view language memory bank, our approach significantly enhances the reconstruction quality and inference speed in sparse-view scenarios.

**Limitations and Prospects.** Our model relies on pre-trained CLIP and SAM models, which limits real-time performance. Moving forward, two key directions are: improving the inference speed of these pre-trained models and developing an integrated approach to directly infer 3D semantic fields from multi-view RGB images, eliminating the need for external models and streamlining the process.

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

# A APPENDIX

## A.1 VIDEO DEMO

In previous experiments, we have shown the visualized results of the feature fields inferred by our method on the RealEstate10K dataset. We provide additional videos showcasing the semantic feature fields and corresponding RGB scenes on the RealEstate10K dataset at https://youtu.be/WfiXiUCKRQ4. Additionally, we present video results of our method on the LERF dataset scenes and compare the feature field results of our method with LangSplat on the 3D-OVS dataset in the end of the video. We highly recommend watching our video demonstrations, as they illustrate the high-quality semantic field representations our method achieves under sparse-view conditions.

## A.2 MORE RESULTS ON LERF DATASET

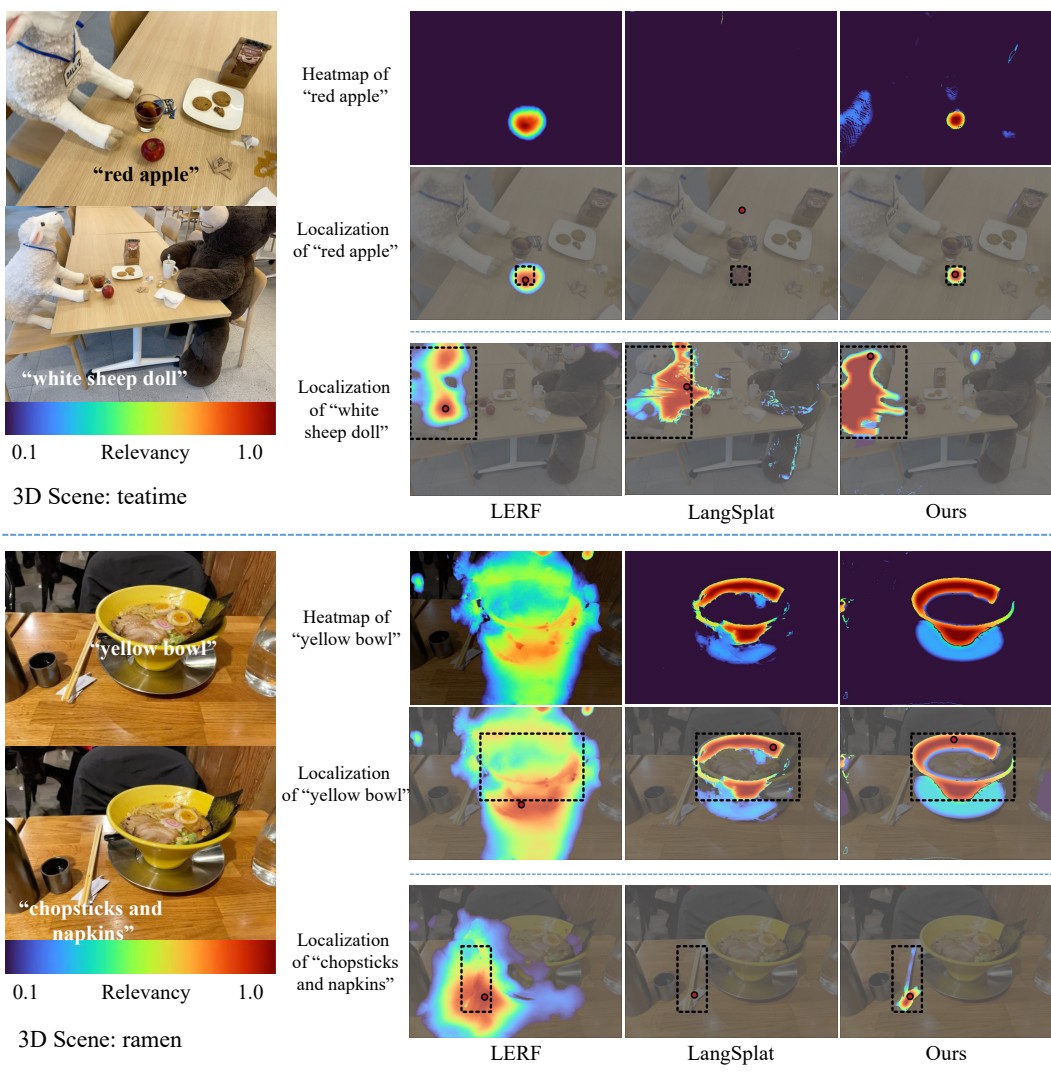

Figure 7: Supplementary results on the LERF dataset. The figure shows open-vocabulary query results for the "teatime" and "ramen" scenes in the LERF dataset.

Figure 7 presents comparative results between our method, LangSplat, and LERF on the LERF dataset. In both the teatime and ramen scenes, LERF struggles to define clear boundaries in 3D

object queries. LangSplat, due to the lack of multi-view information under sparse viewpoint conditions, fails to accurately query the "red apple" and "chopsticks and napkins". For queries like "white sheep doll" and "yellow bowl", LangSplat shows significant issues with missing geometry and object completeness. In contrast, our method produces more complete 3D query results.

## A.3 MORE RESULTS ON 3D-OVS DATASET

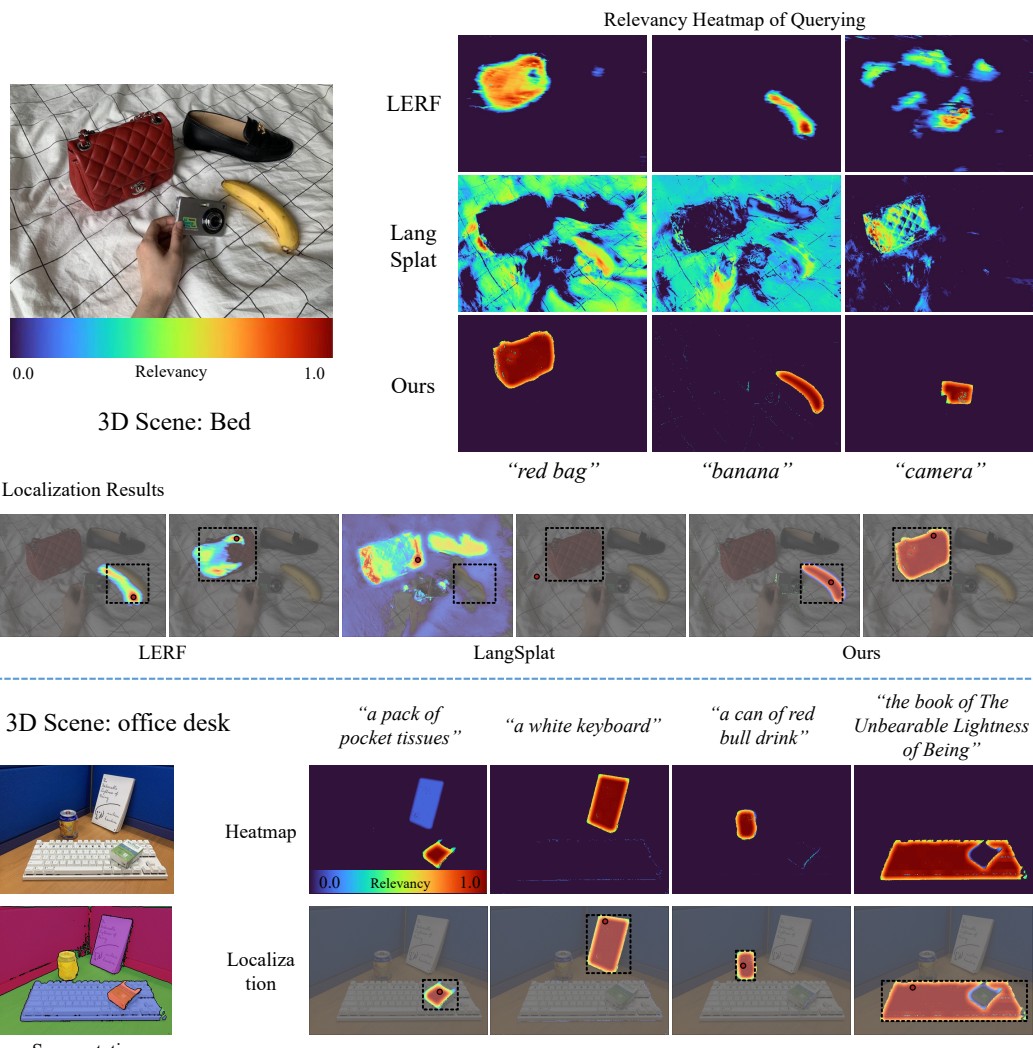

Figure 8: Supplementary results on the 3D-OVS dataset. The figure shows open-vocabulary query results for the "Bed" and "office desk" scenes in the 3D-OVS dataset. In the "Bed" scene, a comparison is provided between the query relevance heatmaps and the localization results of our method, LERF, and LangSplat.

|  | office desk | room | snacks | sofa | table | overall |
|---|---|---|---|---|---|---|
| LERF | 24.08 | 21.77 | 14.56 | 17.89 | 21.04 | 19.87 |
| LangSplat | 22.95 | 40.54 | 25.04 | 28.37 | 51.71 | 33.72 |
| SparseLGS | **64.78** | **59.28** | **58.30** | **46.77** | **59.35** | **57.70** |

Table 7: More quantitative comparisons of 3D semantic segmentation on 3D-OVS dataset. We report the average IoU scores (%).

|  | office desk | room | snacks | sofa | table | overall |
|---|---|---|---|---|---|---|
| LERF | 21.02 | 14.15 | 22.17 | 24.70 | 32.08 | 22.82 |
| 3D-OVS | 26.42 | 14.40 | 16.16 | 20.50 | 18.56 | 19.21 |
| LangSplat | 25.50 | **24.28** | 25.53 | 8.89 | **39.91** | 24.82 |
| SparseLGS | **72.40** | 15.32 | **71.91** | **42.43** | 34.54 | **47.32** |

Table 8: More quantitative comparisons of 3D semantic segmentation on 3D-OVS dataset. We report the average mIoU scores (%).

Figure 8 visualizes the experimental results from the "Bed" and "office desk" scenes in the 3D-OVS dataset. In the "Bed" scene, we compare our method with LangSplat and LERF for 3D object query localization using the query texts "red bag," "banana," and "camera" from a novel viewpoint. As shown in the relevance heatmaps, our method successfully separates the queried objects from the scene with well-preserved object-specific language information. In contrast, LERF and LangSplat struggle to provide coherent scene representations under sparse view conditions. LERF fails to learn sufficient language fields, while LangSplat suffers from misaligned 2D segmentation supervision, causing semantic information to transfer to other objects, leading to incorrect localization. Additionally, we present different query results in the "office desk" scene and the corresponding segmentation results based on the relevance heatmaps. Overall, despite the sparse input of only two viewpoints, our method maintains robust performance in open-vocabulary querying.

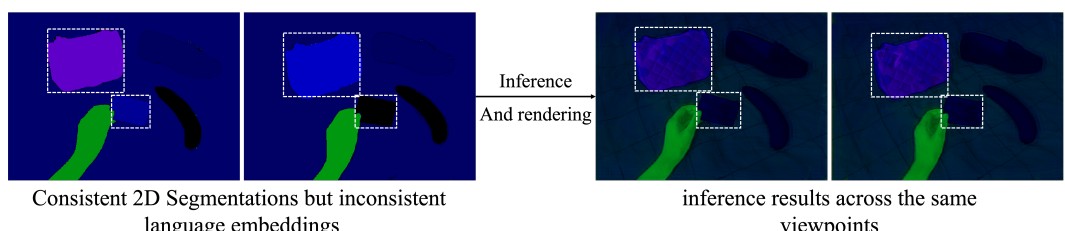

Consistent 2D Segmentations but inconsistent language embeddings

Inference And rendering

inference results across the same viewpoints

Figure 9: Comparison of feature map inputs and rendered results from the same viewpoints without using the Multi-view Language Memory Bank.

### A.4    More Ablation Study Results on RealEstate10K Dataset

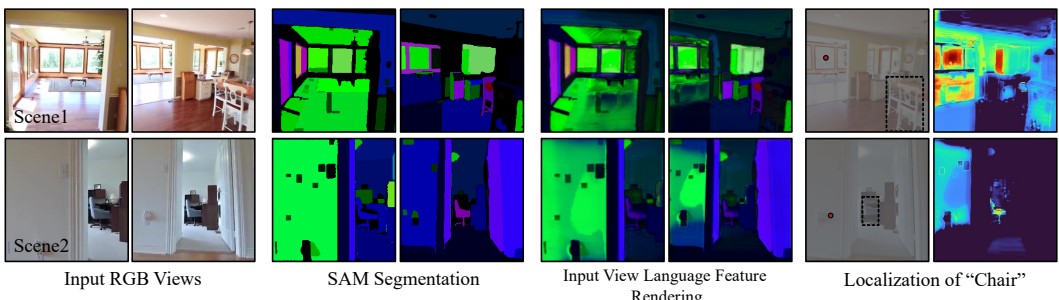

Input RGB Views          SAM Segmentation          Input View Language Feature Rendering          Localization of "Chair"

Figure 10: **Results with inconsistent multi-view segmentation input.** When using only MVSplat and our pre-trained semantic parameters prediction module for semantic field inference on RealEstate10K scenes, the inconsistent multi-view SAM segmentation information results in scrambled semantic field parameters after being decoded by the autoencoder, leading to the inability to accurately query the corresponding objects.

We also conducted ablation experiments on the RealEstate10K dataset, selecting a total of 8 scenes from the test set for evaluation. First, we tested replacing LangSplat's Gaussian optimization process with our proposed Feed-forward Model, transforming it into a similar feed-forward inference

| Component | | | Performance | |
|---|---|---|---|---|
| Feed-forward Model | MA | MV-LMB | Chosen IoU (%) | Localization Accuracy (%) |
| ✓ | | | 16.99 | 31.15 |
| ✓ | ✓ | | 14.76 | 17.38 |
| ✓ | ✓ | ✓ | **41.11** | **64.95** |

Table 9: The ablation study on the RealEstate10K dataset reports Chosen IoU (accuracy of selected object regions) and Localization Accuracy (precision of object localization). The configurations tested include whether the Gaussian feed-forward inference model was used, indicated as Feed-forward Model; whether mask association was performed (MA); and whether the Multi-view Language Memory Bank was performed (MV-LMB).

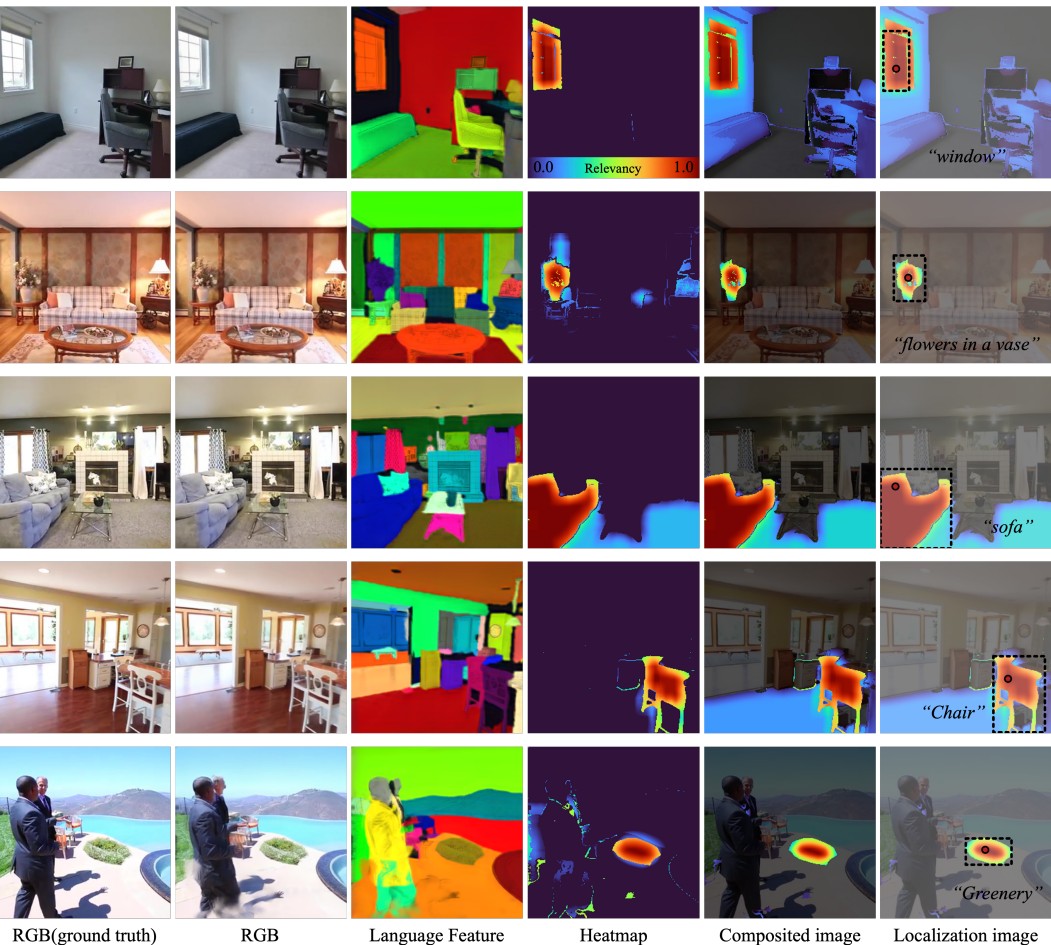

Figure 11: **Additional results of SparseLGS on the re10k test set.** The images above, from left to right, show the ground truth RGB image, predicted RGB image, predicted view semantic feature map, heatmap computed using semantic feature maps and the corresponding query text with the CLIP model, a combination of the heatmap and the RGB image, and the final localization result. All experimental images on the re10k dataset are at a resolution of 256 × 256.

model. In the selected scenes, the average Chosen IoU for the 3D object querying task was 16.99%, and the average Localization Accuracy was 31.15%. Figure 10 illustrates the results of two scenes, showing how the inconsistency in the 2D segmentation from the SAM model across different viewpoints led to highly disorganized semantic feature maps for both input views, resulting in poor final query outcomes. We then incorporated video tracking to impose consistency constraints on the SAM

model's results, but the performance metrics further decreased, which aligns with our findings on the LERF and 3D-OVS datasets. Although multi-view 2D segmentation consistency was ensured, the semantic feature consistency used for supervision was not guaranteed. For more details, see Figure 9; inconsistent language inputs caused the model to fail at recovering even the language information from the input views. Next, we introduced the multi-view language memory bank, which improved the Chosen IoU and Localization Accuracy for 3D object queries to 41.11% and 64.95%, respectively. Figure 11 visualizes the model's inference results for several test set scenes, demonstrating how the introduction of the multi-view language memory bank mapped the 2D segmented view masks into distinct 3D color regions, achieving relatively satisfactory open-vocabulary queries in novel viewpoints.

