# OpenReview forum: "SparseLGS: Fast Language Gaussian Splatting from Sparse Multi-View Images"
_ICLR.cc/2025/Conference — ICLR 2025 Conference Withdrawn Submission_

### Official Review · Reviewer_hdcE · 2024-11-03

**Soundness:** 2
**Presentation:** 3
**Contribution:** 2
**Rating:** 5
**Confidence:** 5

**Summary:**

This manuscript proposes a method for sparse-view open-vocabulary 3D object segmentation and localization based on forward Gaussian representation. The sparse-view capability stems from MVSplat. The association between two sparse-view SAM masks is achieved through video-based tracking. Finally, by establishing a 3D ID and 512D CLIP feature, the Gaussian representation is endowed with open-vocabulary capability.

**Strengths:**

+ This method achieved SOTA performance under sparse-view configurations.
+ In writing, easy to follow.

**Weaknesses:**

1. Method: The method is incremental and combinatorial. (1) The capability of sparse perspective is entirely derived from MVSplat, and I do not believe this is a contribution of SparseLGS. (2) Mask association employs video-based tracking methods that are widely used, such as GaussianGroup, and this approach is not optimal.

2. Fairness of comparison: You provided a YouTube external link video that compared with LangSplat. I found that from the perspective of RGB image rendering, LangSplat performs poorly in novel views, while the high-quality RGB of SparseLGS comes from MVSplat. In LangSplat, appearance and semantics are coupled, and the low-quality appearance also leads to low-quality semantics. Therefore, I believe this comparison is unfair, and your lead over LangSplat may come from MVSplat, which is not your own contribution.

3. Comprehensiveness of comparison: Only one Gaussian-based work was used for comparison, the comparison is weak. There are many open-source Gaussian-based methods that can be used for comparison, such as LEGaussians, GaussianGroup.

4. The experimental setup is unclear and lacks details: the metrics reported in the paper are inconsistent with the official metrics of the comparison methods. However, I couldn’t find any details about how the new metrics were obtained. I can only speculate that they were re-measured on two views.

5. The ablation study (Table 5) provides insufficient information. For example, the performance when using the MA and MV-LMB modules without MVSplat(Feed-forward Model).

6. Table 6, which compares the time efficiency with NeRF-based methods, is unnecessary. Your speed advantage is essentially due to the Gaussian, which has already been demonstrated in previous methods.

**Questions:**

1. Is the number of objects N in each scene manually set? What is the impact of the accuracy of this parameter on the results?

2. In ablation study Table 5, what is your base configuration? In the other configurations without MA and MV-LMB modules, how are the metrics calculated?

3. In Figure 7, you present an example using "chopsticks and napkins" as a text query. I am curious to know what the results would be if chopsticks or napkins were used as queries individually. Based on the video you provided, it seems that you treated these two objects as one. What is the reason for this? Is it due to the setting of N mentioned before?

4. In the Semantic Parameters Prediction module, why is it necessary to train a Res Block and MLP to obtain the three-dimensional features? Why can’t directly use the color of the RGB space divided into N equal parts as the three-dimensional features of the object?

and Weaknesses 2, 4, 5.

---

### Official Review · Reviewer_8hNr · 2024-11-04

**Soundness:** 3
**Presentation:** 3
**Contribution:** 2
**Rating:** 3
**Confidence:** 4

**Summary:**

The paper proposed a new pipeline that can take in only two-view inputs and reconstruct 3D semantic scene (language Gaussians) with a feed-forward 3DGS pipeline (MVSplat). It also relies on "multi-view language bank" to efficiently query semantic features b linking semantic masks to language features. The authors evaluated the proposed method on sparse-view 3D object localization and segmentation to demonstrate the efficacy of the proposed pipeline.

**Strengths:**

1) integrated sparse-view gaussian splatting, LangSplat, SAM and video object tracking.
2) "language memory bank" improves query efficiency.
3) evaluation on 3D-OVS shows promising results.

**Weaknesses:**

1) the technical contributions of this submission is rather limited - it basically integrated MVSplat, SAM, video object tracking and LangSplat. The only novel part is the memory bank which IMO is an incremental improvement to reduce computation complexity in query.
2) the evaluation session did not compare to recent methods such as:
[1] OpenGaussian: Towards Point-Level 3D Gaussian-based Open Vocabulary Understanding
[2] SplatLoc: 3D Gaussian Splatting-based Visual Localization for Augmented Reality
[3] FastLGS: Speeding up Language Embedded Gaussians with Feature Grid Mapping
3) The ablation study is rather confusing - adding feed-forward model in LangSplat degraded the performance so drastically. I don't see how that is entirely from "semantic feature inconsistencies". Even if it is the case, it is straightforward  to get around it (e.g. first construct 3D then assign semantic features like LangSplat does). Adding Mask Association should address the "inconsistent semantic feature" issues by design, but the results became even worse. It is unclear how the authors conducted these experiments.

**Questions:**

I'd like the authors to clarify the above weaknesses especially the novelty. I'd appreciate more detailed explanations w.r.t. the ablation study results. I feel the experiments were not properly designed and the results are insufficiently discussed in the submission hence I found it unconvincing and difficult to understand.

---

### Official Review · Reviewer_DjpE · 2024-11-04

**Soundness:** 3
**Presentation:** 3
**Contribution:** 3
**Rating:** 5
**Confidence:** 3

**Summary:**

In 3D semantic field learning, this paper proposes SparseLGS, a feed-forward method for constructing 3D semantic fields from sparse viewpoints, allowing direct inference of 3DGS-based scenes.
The method surpasses state-of-the-art approaches in sparse-view 3D object localization and segmentation tasks with faster speed.

**Strengths:**

1. This method is the first feed-forward method in 3D semantic field learning. It is more convenient in practice than per-scene multi-view optimizations.
2. The performance and speed of the method are excellent.
3. The  quantitative results are good.

**Weaknesses:**

1. In the experiment, how many views are used to train the contrastive method? It seems that the performance of LangSplat is much lower than the results reported in the original paper.
2. This method rely on a video tracking models. It would be best if the author could verify the robustness of the method to the tracking model effect, and whether incorrect tracking would seriously affect the final performance.

**Questions:**

Is this method able to support more pictures as input to obtain more multi-view information ? It seems difficult to learn semantic fields with only two view.

---

### Official Review · Reviewer_nrH6 · 2024-11-04

**Soundness:** 3
**Presentation:** 3
**Contribution:** 2
**Rating:** 5
**Confidence:** 3

**Summary:**

This paper proposes a feed-forward method that uses only a few viewpoints to construct 3D semantic fields. SAM segmentations are used for consistent video tracking and CLIP features are used to embed language information in 3D space for open-vocabulary capability. Experimental performance show that the proposes method outperforms existing approaches in the LERF and 3D-OVS datasets.

**Strengths:**

This is a well-written and easy to follow paper with impressive experimental results.

**Weaknesses:**

My main concern is regarding the limited technical novelty. This paper combines different existing approaches and stitches them together to obtain good results. Feed-forward approaches, integrating language through CLIP features, using SAM segmentations, etc, are all concepts that have been explored before. The multi-view language memory bank that links semantic masks to natural language information has been explored extensively in CLIP-based segmentation networks (eg., [1]).

[1] Image Segmentation Using Text and Image Prompts, Lüddecke et al., CVPR 2022

**Questions:**

1. The authors report that 2D Query achieves a higher Chosen IOU as compared to the proposed method in table 6. Some intuition on why that is the case would be useful.

---

> ### Author Response · Authors · 2024-11-13
>
> Thank you so much for your thoughtful review and the kind words about our paper’s readability and experimental results. We genuinely appreciate your insights, and we’d like to address your main concerns and questions.
>
> **Technical Novelty and Relevance to Suggested Literature**
>
> We understand your concern regarding the technical novelty, especially in light of the paper you referenced: *Image Segmentation Using Text and Image Prompts* by Lüddicke et al. (CVPR 2022). After carefully reviewing this work, we found that it focuses on 2D image segmentation tasks, specifically integrating semantic segmentation masks with natural language information. In contrast, our work is focused on 3D semantic field construction and open-vocabulary querying within 3D space, which introduces unique challenges and contributions not addressed in 2D segmentation.
>
> Our multi-view language memory bank is particularly designed to aggregate multi-view language information and store it within a 3D scene representation. This enables robust open-vocabulary semantic querying from novel viewpoints, a task that requires spatial consistency across views in 3D, rather than simply associating language with 2D segmentation. We hope this clarifies the distinct contributions and scope of our work compared to existing literature.
>
> **2D Query Performance Compared to 3D Query Results**
>
> Regarding your question about why our 2D query achieves a higher Chosen IoU than our 3D query results, this can be attributed to the inherent complexities in integrating 2D semantic information into a 3D scene. When we embed 2D semantic information into the 3D scene representation, some information loss is inevitable, which may slightly impact the fidelity of the final 3D semantic output compared to the initial 2D input.
>
> For instance, when directly using rendered RGB views from novel perspectives in our 3D model and processing them through the SAM and CLIP model, we sometimes achieve slightly higher performance metrics due to the preservation of high-fidelity visual details in 2D. However, as shown in our tables, this 2D processing pipeline is significantly slower, taking up to around 18 seconds per query, which is impractical for real-time applications. In contrast, our 3D method achieves query speeds around 0.011 seconds, making it far more suitable for interactive 3D spatial applications, where fast response times are critical.
>
> **Future Directions**
>
> We acknowledge the reliance on pre-trained CLIP and SAM models, and we see this as an area with room for further improvement. Moving forward, we are keen to explore developing a more integrated model that does not depend on external components, allowing for direct inference from multi-view RGB images. Additionally, enhancing the inference speed of these components could help us approach real-time performance.
>
> We would also like to clarify that while our approach leverages existing feed-forward models, CLIP features, and SAM segmentation to perform 3D semantic field inference from sparse viewpoints, we have combined the strengths of these models to propose a novel paradigm for 3D semantic field learning. Our method departs from traditional approaches that involve separate steps of scene reconstruction followed by 3D semantic learning in a per-view, per-scene manner, which often takes several minutes. Instead, we offer a more streamlined solution for sparse-view scenarios, enabling efficient 3D semantic inference without the lengthy per-view optimization process. We hope this approach highlights our contribution to advancing 3D semantic field learning in a more efficient and effective way.
>
> Thank you once again for your invaluable feedback. We hope these clarifications address your concerns and provide a clearer understanding of our contributions.

---

### Note · Authors · 2024-11-14

I have read and agree with the venue's withdrawal policy on behalf of myself and my co-authors.